# Pancarditis as the Clinical Presentation of Eosinophilic Granulomatosis with Polyangiitis: A Multimodality Approach to Diagnosis

Michele Lioncino [1,†], Emanuele Monda [1,†], Santo Dellegrottaglie [2,3], Annapaola Cirillo [1], Martina Caiazza [1], Adelaide Fusco [1], Francesca Esposito [4], Federica Verrillo [1], Giovanni Ciccarelli [5], Marta Rubino [1], Massimo Triggiani [6], Raffaele Scarpa [7], Alida Linda Patrizia Caforio [8], Renzo Marcolongo [9], Stefania Rizzo [10], Cristina Basso [10], Gerardo Nigro [5], Maria Giovanna Russo [11], Paolo Golino [5] and Giuseppe Limongelli [1,*]

1   Inherited and Rare Cardiovascular Disease Unit, AORN dei Colli, Monaldi Hospital, 80131 Naples, Italy; michelelioncino@icloud.com (M.L.); emanuelemonda@me.com (E.M.); cirilloannapaola@gmail.com (A.C.); martina.caiazza@yahoo.it (M.C.); adelaidefusco@hotmail.it (A.F.); fedeverrillo@gmail.com (F.V.); rubinomarta@libero.it (M.R.)
2   Division of Cardiology, Villa dei Fiori, 80011 Acerra, Italy; sandel74@hotmail.com
3   Mount Sinai School of Medicine, New York, NY 10029, USA
4   Emergency Division, AORN Moscati, 83100 Avellino, Italy; francescaesposito84@yahoo.it
5   Department of Cardiology, University 'L Vanvitelli'—Monaldi Hospital, 80131 Naples, Italy; ciccarelli.giovanni@gmail.com (G.C.); gerardo.nigro@unicampania.it (G.N.); paolo.golino@unicampania.it (P.G.)
6   Division of Allergy and Clinical Immunology, Department of Medicine, University of Salerno, 84121 Salerno, Italy; massimo.triggiani@sangiovannieruggi.it
7   Rheumatology Unit, Department of Clinical Medicine and Surgery, University Federico II, 80131 Naples, Italy; rscarpa@unina.it
8   Cardiology, Department of Cardiac, Thoracic, Vascular Sciences and Public Health, University of Padua, 35128 Padua, Italy; alida.caforio@unipd.it
9   Hematology and Clinical Immunology, Department of Medicine, University of Padova, 35128 Padova, Italy; renzo.marcolongo@aopd.veneto.it
10  Cardiovascular Pathology, Department of Cardiac, Thoracic, Vascular Sciences and Public Health, University of Padova, 35128 Padova, Italy; s.rizzo@unipd.it (S.R.); cristina.basso@unipd.it (C.B.)
11  Pediatric Cardiology, University 'L Vanvitelli'—Monaldi Hospital, 8013 Naples, Italy; mgiovannarusso@gmail.com
*   Correspondence: limongelligiuseppe@libero.it
†   These authors contributed equally to this work.

**Abstract:** Eosinophilic pancarditis (EP) is a rare, often unrecognized condition caused by endomyocardial infiltration of eosinophil granulocytes (referred as eosinophilic myocarditis, EM) associated with pericardial involvement. EM has a variable clinical presentation, ranging from asymptomatic cases to acute cardiogenic shock requiring mechanical circulatory support (MCS) or chronic restrictive cardiomyopathy at high risk of progression to dilated cardiomyopathy (DCM). EP is associated with high in-hospital mortality, particularly when associated to endomyocardial thrombosis, coronary arteries vasculitis or severe left ventricular systolic dysfunction. To date, there is a lack of consensus about the optimal diagnostic algorithm and clinical management of patients with biopsy-proven EP. The differential diagnosis includes hypersensitivity myocarditis, eosinophil granulomatosis with polyangiitis (EGPA), hypereosinophilic syndrome, parasitic infections, pregnancy-related hypereosinophilia, malignancies, drug overdose (particularly clozapine) and Omenn syndrome (OMIM 603554). To our knowledge, we report the first case of pancarditis associated to eosinophilic granulomatosis with polyangiitis (EGPA) with negative anti-neutrophil cytoplasmic antibodies (ANCA). Treatment with steroids and azathioprine was promptly started. Six months later, the patient developed a relapse: treatment with subcutaneous mepolizumab was added on the top of standard therapy, with prompt disease activity remission. This case highlights the role of a multimodality approach for the diagnosis of cardiac involvement associated to systemic immune disorders.

**Keywords:** eosinophilic myocarditis; eosinophilic granulomatosis with polyangiitis; pancarditis

## 1. Introduction

Eosinophilic pancarditis (EP) is a rare, often unrecognized condition caused by myocardial infiltration of eosinophil granulocytes (referred as eosinophilic myocarditis, EM) associated with pericardial and endocardial involvement. EM has a variable clinical presentation, ranging from asymptomatic cases to acute cardiogenic shock requiring mechanical circulatory support (MCS) or chronic restrictive cardiomyopathy at high risk of progression to dilated cardiomyopathy (DCM) [1–6].

While the exact prevalence of EP is unknown, EM has been detected in up to 0.5% of unselected autopsy series and in more than 20% of explanted hearts from heart transplant recipients after hypersensitivity-mediated acute rejection, although there are not systematic large studies addressing the real prevalence of EM [7–9]. Peripheral eosinophilia often accompanies EM, being detected in about 70% of the cases, although the severity of eosinophilia seems not related to the severity of disease [1,10]. To date, endomyocardial biopsy (EMB) is required to achieve a definite diagnosis [11], and carries a IIa recommendation according to the American Heart Association/American College of Cardiology (AHA/ACC) statement consensus, particularly in life-threatening conditions [12], whilst the 2013 European Society of Cardiology (ESC) consensus statement recommends to perform EMB early in the course of disease to achieve diagnosis and for prognostic stratification [13]. In addition, another ESC consensus recommends EMB in patients with clinically suspected myocarditis in the context of systemic immune-mediated diseases (SIDs) [14].

EP is associated to high in-hospital mortality, particularly in patients showing endocardial thrombosis, coronary arteries vasculitis or severe left ventricular systolic dysfunction [3,15,16]. There is conflicting evidence about the optimal diagnostic work-up and therapeutic management of patients with EP.

To our knowledge, herein we report the first case of pancarditis associated to eosinophilic granulomatosis with polyangiitis (EGPA) with negative anti-neutrophil cytoplasmic antibodies (ANCA), showing a multimodal approach to diagnosis.

## 2. Case Presentation

A 40-year old woman was referred to our unit from a primary healthcare hospital with a clinically suspected myocarditis. The patient had been in her usual state of health until three weeks before, when atypical chest pain appeared. She reported a family history of immune-mediated/autoimmune disease: her father was suffering for psoriatic arthritis and her mother from type I diabetes mellitus. Seven years before the evaluation, the patient had experienced an episode of pericarditis following an acute respiratory infection, which resolved after treatment with aspirin and low-dose colchicine.

She had a medical history of severe asthma with forced expiratory volume in the 1st second (FEV1) < 50% of the predicted value and three years before she underwent surgery for nasal polyposis. She denied any cardiovascular risk factors, as well as smoking, alcohol assumption or any illicit drug injection. Three months before, she received the second dose of the BNT121b1 mRNA SARS-CoV2 vaccine. The patient did not report any history of fever, night sweats, unintentional weight loss, dyspnea, gastrointestinal symptoms, joint pain, myalgia or cutaneous manifestations. She reported intermittent use of non-steroidal anti-inflammatory drugs (NSAIDs) for recurrent episodes of headache. Laboratory tests, performed at Emergency department, showed a mild increase of high sensitivity cardiac troponin (hs-TnI 48 ng/L, 99th percentile upper reference limit 14 ng/L) and increased eosinophil count (2080/mm$^3$, 19.5%, total blood cell count $10.68 \times 10^3$/microL). Serum creatinine, thyroid enzymes, electrolytes, and serum transaminases were within the reference ranges. C-reactive protein, platelet count and ferritin were increased (respectively: 8 mg/dL, $300 \times 10^3$/microL and 420 micrograms/L). A coronary angiography was then

performed showing normal coronary anatomy without significant coronary atherosclerosis. In the suspicion of myopericarditis, cardiac magnetic resonance (CMR) with a contrast medium was performed. Overall, late-gadolinium enhancement images were suggestive of an active inflammatory process simultaneously affecting the endocardium (with a prominent involvement of the left ventricular papillary muscles and mitral leaflets), the myocardium of the left and right ventricles and the left and right atria, and the pericardial layers [17] (the so-called "pancarditis"; Figure 1).

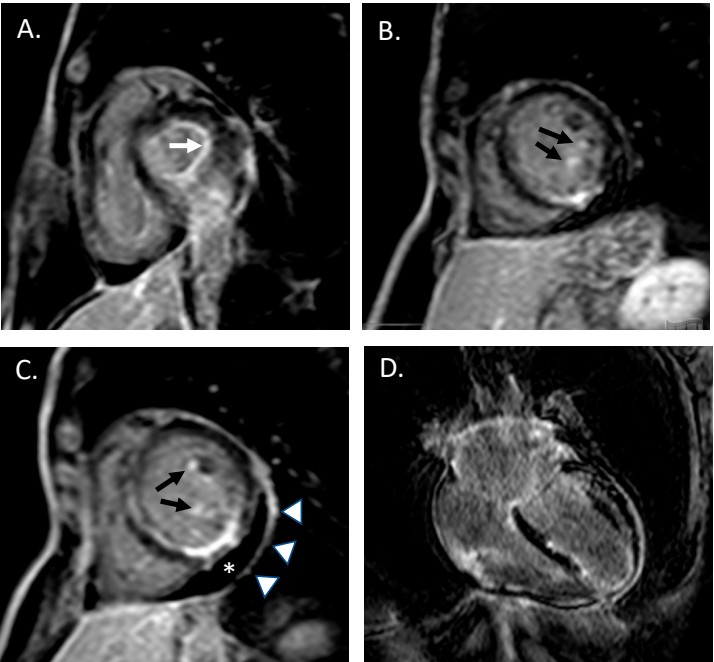

**Figure 1.** Cardiac MRI acquisitions obtained late after contrast injection, showing multiple areas of late gadolinium enhancement (LGE, "bright signal") simultaneously affecting the endocardium, the myocardium and the pericardium. (**A**) short-axis view images documenting LGE involvement of mitral valve anulus (white arrow) and (**B**,**C**) left ventricular papillary muscles; (**C**) short-axis mid papillary view showing mild pericardial effusion (asterix) and pericardial enhancement; (**D**) four-chamber view image showing diffuse myocardial LGE involving left and right ventricles and both atria.

At our Unit, the patient appeared well, without signs of dyspnea or diaphoresis. Her pulse was 73 beats/minute, blood pressure was 110/70 mmHg, respiratory rate was 18/minute, and hemoglobin saturation was 99% while she was breathing ambient air. A 12-lead ECG was performed, showing repolarization abnormalities in inferior leads without PR depression or other signs of pericarditis (Figure 2). Extended microbiological screening, comprehensive of cultures of blood and stool for ova parasites and serum antibodies for *Strongyloides* spp. excluded parasitosis. Anti-neutrophil cytoplasmic antibodies (ANCA) and antinuclear antibodies (ANA) were absent. A real time polymerase chain reaction (RT-PCR) nasopharyngeal swab for SARS CoV 2 ruled out Coronavirus infection.

Advanced 2D echocardiogram showed normal systolic function with a mild reduction of left ventricular global longitudinal strain (GLS—14.5%, GE Healthcare Software, Little Chalfont, UK). A small circumferential pericardial effusion (<10 mm) was detected in the parasternal short-axis view (Figure 3).

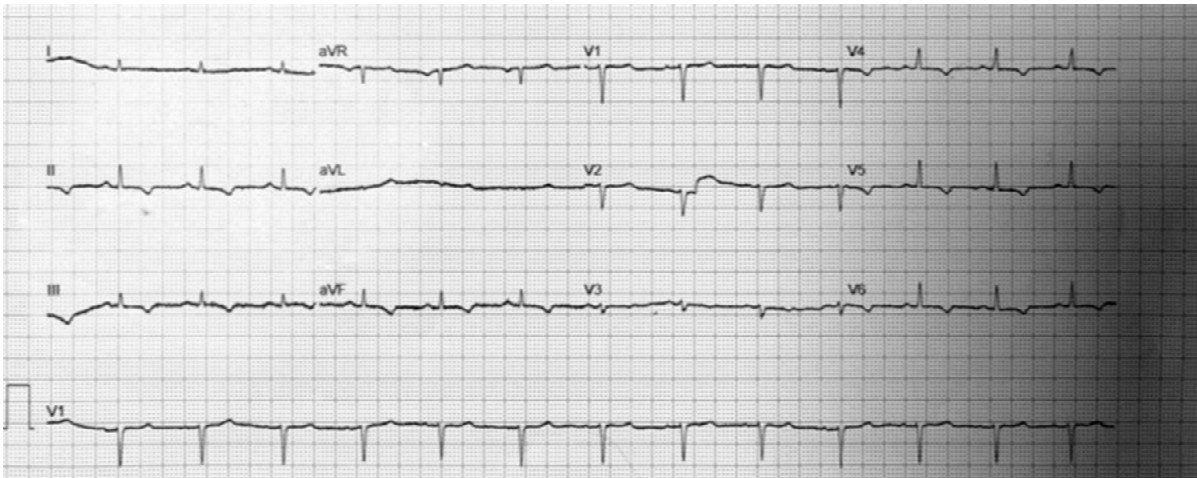

**Figure 2.** 12-lead ECG showing sinus rhythm and repolarization abnormalities in inferior leads, in absence of PR interval depression or other signs of pericarditis.

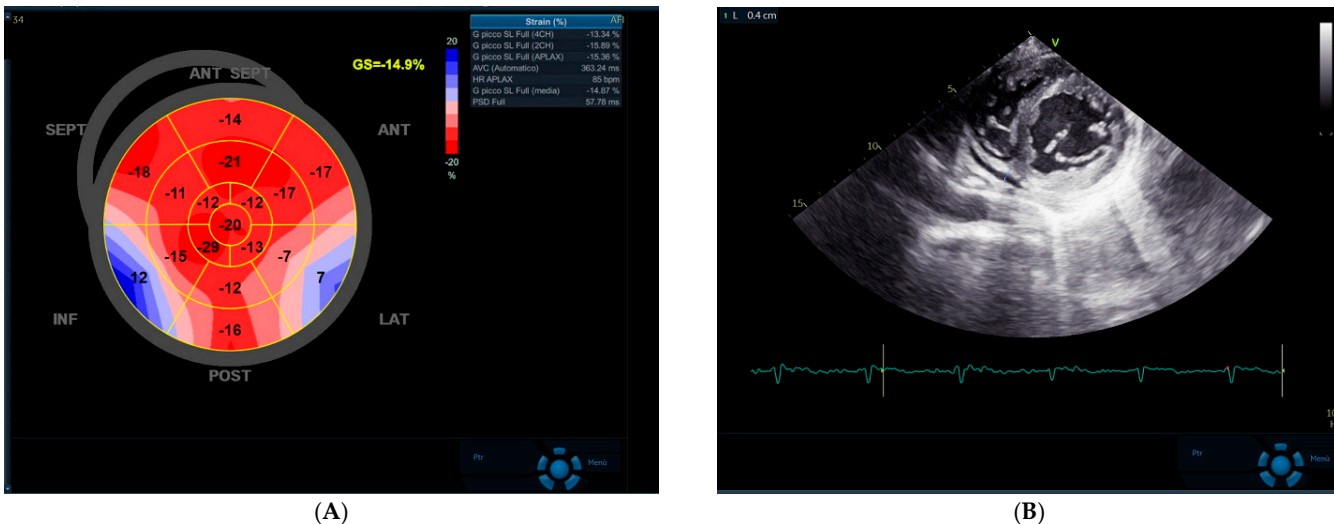

(**A**)                                                                                     (**B**)

**Figure 3.** 2D-Echocardiogram showing small reduction in left ventricular global longitudinal strain (**A**) and mild, circumferential pericardial effusion (**B**).

For the purposes of diagnostic confirmation, the patient underwent EMB. Histopathological findings included interstitial and perivasal myocardial infiltrates, mainly composed by lymphocytes, granulocytes and scattered eosinophils (Figure 4).

A multidisciplinary teleconsulting heart team involving different specialists, including cardiologists, rheumatologists, immunologists, pathologists, and geneticists was designated to evaluate the best management options. Although the patient fulfilled the 1990 American College of Rheumatology criteria (ACR) for the diagnosis of EGPA [18], the exclusion of an underlying hypereosinophilic syndrome (HES) and of a hematologic malignancy was mandatory.

Whole body positron emission tomography (PET) revealed mildly increased 18-fluorodeoxyglucose uptake (18-FDG) in mediastinal lymph nodes (maximal standardized uptake value SUV 2.63) consistent with diffuse nonspecific inflammatory status (Figure 5).

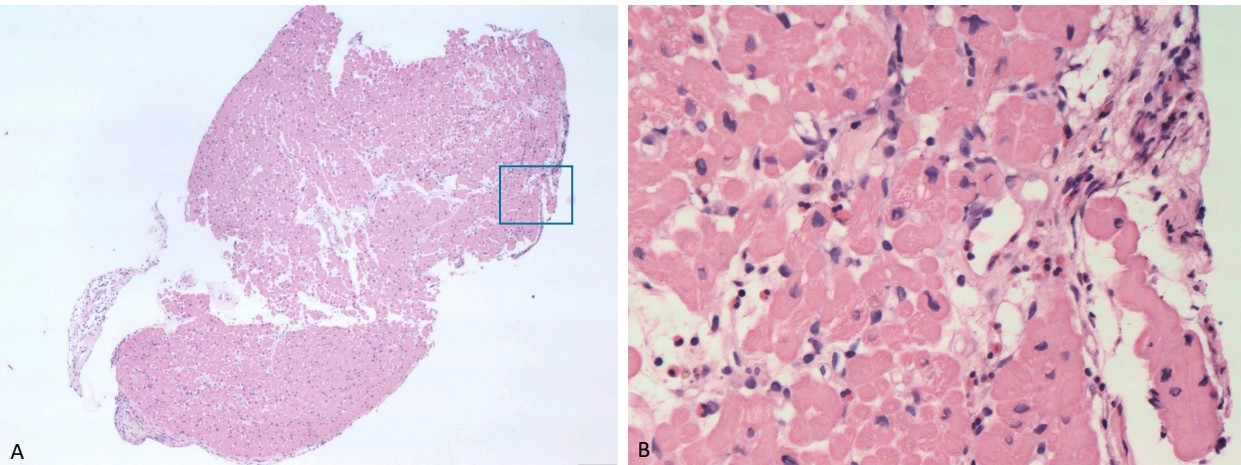

**Figure 4.** (**A**) Endomyocardial biopsy specimen, showing at higher magnification (**B**) an interstitial inflammatory infiltrate with eosinophils (Haematoxylin and eosin, scale bars 200 micron (**A**) and 50 micron (**B**).

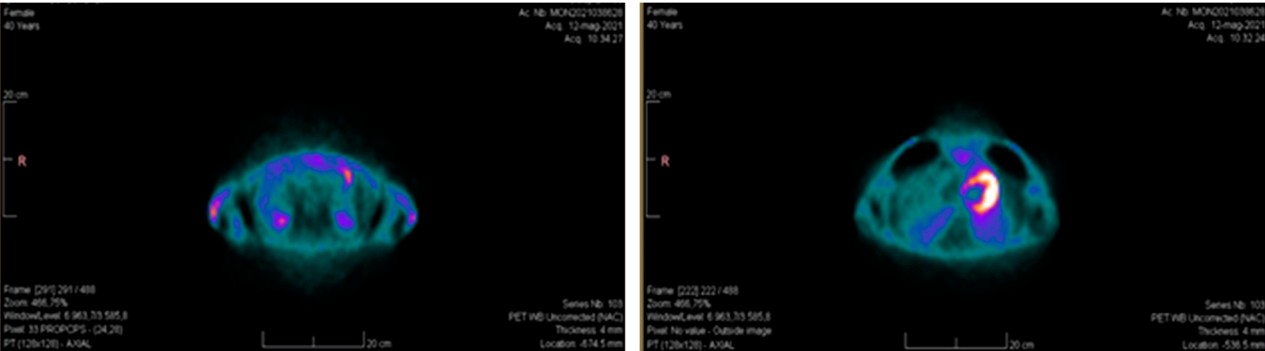

**Figure 5.** Whole-body positron emission tomography (PET) showing mildly increased 18-fluorodeoxyglucose uptake (18-FDG) in mediastinal lymph nodes (maximal standardized uptake value SUV 2.63), consistent with diffuse nonspecific inflammatory status.

Fluorescence-in situ-hybridization (FISH) excluded F1P1L1-PGFRα gene rearrangement and, at karyotype analysis, recurrent translocations associated to myeloproliferative variants of hypereosinophilia were not detected.

Treatment with prednisone (50 mg/day) was promptly started to induce clinical remission and tapered to 10 mg/day within three months. As the patient declined treatment with cyclophosphamide, genetic polymorphisms for thiopurine-methyltranspherase were excluded and azathioprine (2 mg/kg day body weight BW) was added to prednisone as a steroid sparing drug. Remission, defined as Birmingham Vasculitis Activity Score (version 3) = 0, was achieved by the combination of low-dose prednisone (10 mg/d) and azathioprine. Six months later, the patient developed a relapse, after which she was treated with prednisone and azathioprine, defined as a BVAS > 3 associated with asthma worsening. Consequently, treatment with subcutaneous mepolizumab (300 mg monthly) was added on the top of standard immunosuppressive therapy, with prompt disease activity remission [19]. After four months follow-up, the patient is in good clinical state and has not experienced any relapse.

## 3. Discussion

There is a lack of consensus about the optimal diagnostic algorithm and clinical management of patients with biopsy-proven EP [1,7,12], although a recent consensus underlies the central role of EMB for differential diagnosis [14]. Differential diagnosis includes

hypersensitivity myocarditis, EGPA, hypereosinophilic syndrome, parasitic infections, pregnancy-related hypereosinophilia, malignancies, drug overdose (particularly clozapine) and Omenn syndrome (OMIM 603554).

Hypersensitivity myocarditis is the most common etiology of EM, and it is associated to the highest mortality rates (about 30%) [1], although possibly over-estimated due to publication bias, for the presence of non-biopsy proven EM with mild presentation having a lower chance of being reported. Pericardial involvement has been reported in 30% of patients with hypersensitivity-mediated EM [1]. Although more than 20 drugs have been reported to be associated with hypersensitivity myocarditis, antitubercular drugs, sulphonamides and β-lactam antibiotics are the most frequently involved [20]. The time interval between drug-exposure and the onset of symptoms may vary between months and hours and active eosinophil infiltrates could persist even after drug discontinuation [20–23]. Notably, vaccines were associated to EM only in 7.7% of the cases [1] and although rare cases have been reported among recipients of SARS CoV 2 BNT162b2 vaccination [24], it is still uncertain whether these are caused by the vaccine, since in the anecdotal biopsy-proven cases viral etiology was not ruled out on EMB [25] and the risk-benefit ratio for vaccination remains unmodified. Although the hypothesis of hypersensitivity myocarditis could be taken into account on the basis of the personal history of frequent NSAIDs assumption, asthma and nasal polyposis are uncommon, whereas cutaneous or systemic manifestations are often reported.

A complete microbiological assessment was performed to exclude parasitic infection, whereas Omenn syndrome was ruled out for the age of symptom presentation, the absence of erythroderma, chronic diarrhea, malabsorption or signs of severe combined immunodeficiency.

Hypereosinophilic syndrome is more common among women and is more commonly associated to peripheral eosinophilia than hypersensitivity reactions [1,4,26]. The World Health Organization's (WHO) 2019 clinical statement [26] has endorsed a molecular classification of hypereosinophilic syndromes. Screening for F1P1L1-PGFRα gene rearrangement by FISH or real time-polymerase chain reaction (PCR) represents the first step in the evaluation of suspected clonal eosinophilia and can identify patients who will be eligible for treatment with imatinib [27]. The absence of the F1P1L1-PGFRα fusion gene should prompt evaluation for recurrent translocations associated to myeloid neoplasms with eosinophilia and FL1P1L1, PGFRα, PDGFRβ and PMC1-JAK2 rearrangements [28]. Our patient showed a normal karyotype and, although the absence of translocations can be rarely associated to a myeloproliferative disorder classified as chronic eosinophilic leukaemia-not otherwise specified (CEL-NOS), the absence of circulating blasts at blood smear and the presence of an alternative diagnosis ruled out this hypothesis [29].

EGPA was then diagnosed according to the ACR criteria [18]. Of note, the absence of ANCA and the 18-FDG uptake pattern could orient the diagnosis towards hematologic malignancies. However, it should be noted that the prevalence of ANCA-positive patients is lower in EGPA than in other small-vessel vasculitides (40%) [30]. In particular, younger age and higher prevalence of refractory asthma and cardiovascular involvement have been reported among ANCA-negative patients [31]. Whole-body 18-FDG PET may not show the typical pattern of increased uptake in the paranasal sinuses in 30% of the cases: mild mediastinal 18-FDG uptake may reflect chronic inflammation rather than hematologic malignancy. Pancarditis is a rare presentation of EGPA, and is associated with a high risk of systolic dysfunction and evolution to a dilated or restrictive cardiomyopathy [5,32]. The history of asthma and nasal polyposis, in association with peripheral eosinophilia, could represent useful red flags of an underlying EGPA.

Because EGPA-associated pancarditis carries a high risk of endocavitary thrombosis, there is a lack of consensus about the need for antithrombotic treatment among these patients. The release of eosinophil major cationic protein has been advocated as a possible mechanism for thrombosis due to its capacity to bind thrombomodulin [33]. The presence of endocavitary thrombus may increase the risk of systemic embolism associated to left-

sided endomyocardial biopsy and careful multimodality imaging should be performed in patients before EMB, in order to exclude embolic sources.

Mepolizumab is an anti-interleukin 5 (IL5) monoclonal antibody, originally developed for severe asthma, which prevents IL5 from binding to its receptor on cellular surface.

To our knowledge, mepolizumab has been employed in combination with rituximab to induce remission in a single case of ANCA-negative EGPA with intractable myocarditis [34]. In particular, mepolizumab was underdosed to 100 mg/monthly during the induction protocol due to its association to rituximab, and then used as monotherapy. According to a double blind randomized controlled trial comparing mepolizumab and placebo [19], we administered standard doses of mepolizumab (300 mg monthly) in association to steroid therapy [35]. Recently, benralizumab has been demonstrated to be well tolerated and effective in the treatment of EGPA-associated EM [36]. In particular, benralizumab binds the alpha subunit of human interleukin-5 receptors and, as a result of antibody-directed cell cytotoxicity, it has increased eosinophil-depleting activity compared with mepolizumab. Further studies are required to assess the role of benralizumab in the treatment of EGPA-associated myocarditis.

The case highlights the role of a multimodality approach for the diagnosis of cardiac involvement associated with systemic immune disorders.

**Author Contributions:** G.L, and E.M. conceived and ideated the manuscript. M.L., G.L., M.C., M.T., R.S., A.L.P.C., R.M., C.B. and S.R. drafted the manuscript. S.D., A.C., A.F., F.E., F.V., G.C., M.R., G.N., M.G.R. and P.G. critically reviewed the manuscript. All authors have read and agreed to the published version of the manuscript.

**Funding:** This research received no external funding.

**Institutional Review Board Statement:** The study was conducted according to the guidelines of the Declaration of Helsinki, and approved by the Institutional Review Board of AORN dei Colli, Monaldi Hospital.

**Informed Consent Statement:** Informed consent was obtained from all subjects involved in the study. Written informed consent has been obtained from the patient(s) to publish this paper.

**Acknowledgments:** We acknowledge Michela Piscopo, Daniela Lafera and Ciro de Prisco.

**Conflicts of Interest:** The authors declare that they have no conflict of interest.

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
