# Peer review of "Pancarditis as the Clinical Presentation of Eosinophilic Granulomatosis with Polyangiitis: A Multimodality Approach to Diagnosis"

_cardiogenetics, doi:10.3390/cardiogenetics12020014_

Round 1

Reviewer 1 Report

In this paper, authors refer to an interesting case of pancarditis associated to eosinophilic granulomatosis with polyangiitis with negative ANCA. The paper is well written and the imaging available is very interesting.

Some suggestions:

1) in the multimodality presentation, I would suggest to add the EKG of the patient;

2) in the Introduction, when authors said that "EM has a variable clinical
presentation, ranging from asymptomatic cases to acute cardiogenic shock requiring mechanical circulatory support", it could be interesting to refere also to "Di Nora C. A case of hypereosinophilic syndrome with Loeffler endocarditis successfully bridged to heart transplantation by biventricular mechanical support. Artif Organs. 2021 Jul;45(7):784-785".

Author Response

In this paper, authors refer to an interesting case of pancarditis associated to eosinophilic granulomatosis with polyangiitis with negative ANCA. The paper is well written and the imaging available is very interesting.

Some suggestions:

1) in the multimodality presentation, I would suggest to add the EKG of the patient;

2) in the Introduction, when authors said that "EM has a variable clinical
presentation, ranging from asymptomatic cases to acute cardiogenic shock requiring mechanical circulatory support", it could be interesting to refere also to "Di Nora C. A case of hypereosinophilic syndrome with Loeffler endocarditis successfully bridged to heart transplantation by biventricular mechanical support. Artif Organs. 2021 Jul;45(7):784-785".

We thank the Reviewer for the interesting comments

- In the multimodality presentation, we added the electrocardiogram of the patient and a brief decription of its interpretation

Figure 2 12-lead ECG showing sinus rhytm and repolarization abnormalities in inferior leads, in absence of PR interval depression or other signs of pericarditis

-In the Introduction section, we added the suggested reference

Reviewer 2 Report

The authors present a very interesting case of eosinophilic pancarditis that appears to be associated with EGPA. They nicely describe the patient's history and course as well as diagnostic and treatment modalities. The approach to diagnosis outlined here would be valuable for others to read. The manuscript would benefit from a few updates, particularly related citations and the treatment response/follow-up, but this is otherwise nicely done.

Comments below organized by section of the manuscript:

INTRO

  • Line 53: Up to 0.5% of unselected autopsy series and more than 20% of explanted hearts from transplant recipients - are these numbers correct? Both of these seem very high, and the only citation for this sentence is a 3 patient case series – need more references for these claims. Also one of the examples seems to describe hypersensitivity mediated acute rejection of a transplanted heart that was explanted - this does not seem like the best patient example for this manuscript, different issue. 

CASE PRESENTATION

  • Line 81 – should say <50% predicted (or of predicted value, something like that)
  • Nice thorough report of key historical factors, including NSAID use
  • Line 89 – may be more appropriate to say “and” rather than “associated with” for the troponin and eos
  • Line 141 – consider “patient declined” rather than “patients refused”
  • Line 144-148
    • Please state whether the patient achieved initial remission (and how that was defined, ? BVAS 0) while on azathioprine and prednisone (sounds like she remained on 10mg/d prednisone).
    • If going to report BVAS score, need to state which version you are using.
    • Great that she achieved remission with introduction of mepolizumab. Please comment on duration of follow-up at the time of manuscript submission.
    • Are there any follow-up imaging studies available to determine whether the MRI or PET findings improved after initiation of therapy? This would certainly be interesting to the audience.

DISCUSSION

  • Line 218 – there is an incomplete sentence starting with “Recently, benralizumab”

AUTHOR CONTRIBUTIONS, IRB STATEMENT, INFORMED CONSENT STATEMENT – these sections have not been completed by the authors

FIGURE 1 – phrasing/sequence of the caption could be clearer. May be clearer to state which image is being described prior to the description.

FIGURE 2 – Consider saying “small” instead of “mild”

Author Response

Reviewer 2

The authors present a very interesting case of eosinophilic pancarditis that appears to be associated with EGPA. They nicely describe the patient's history and course as well as diagnostic and treatment modalities. The approach to diagnosis outlined here would be valuable for others to read. The manuscript would benefit from a few updates, particularly related citations and the treatment response/follow-up, but this is otherwise nicely done.

Comments below organized by section of the manuscript:

INTRO

  • Line 53: Up to 0.5% of unselected autopsy series and more than 20% of explanted hearts from transplant recipients - are these numbers correct? Both of these seem very high, and the only citation for this sentence is a 3 patient case series – need more references for these claims. Also one of the examples seems to describe hypersensitivity mediated acute rejection of a transplanted heart that was explanted - this does not seem like the best patient example for this manuscript, different issue. 

We thank the reviewer for the interesting question and apologize for the incomplete reference. We acknowledge with the reviewer regarding the high prevalence of EM according to the autopsy series. Numerous studies report similar prevalence of eosinophilic infiltration in explanted hearts:

  • https://jmedicalcasereports.biomedcentral.com/articles/10.1186/1752-1947-4-40
  • Baandrup U. Cardiac Pathology. London: Springer; 2013. Myocarditis/inflammatory cardiomyopathy; pp. 133–146
  • Winters G, McManus BM: Myocarditis. Cardiovascular Pathology. Edited by: Silver MD, Gotlieb AI, Schoen FJ. 2001, New York: Churchill Livingstone, 256-3
  • Sheikh H, Siddiqui M, Uddin SMM, Haq A, Yaqoob U. The Clinicopathological Profile of Eosinophilic Myocarditis. Cureus. 2018;10(12):e3677. Published 2018 Dec 3. doi:10.7759/cureus.3677

However, we modified the introduction section as follows

“While the exact prevalence of EP is unknown, EM has been detected in up to 0.5% of unselected autopsy series and  in more than 20% of explanted hearts from heart transplant recipients after hypersensitibity-mediated acute rejection, although there are not systematic large studies addressing the real prevalence of EM”

Furthermore we updated the introduction section adding new references (6-9)

CASE PRESENTATION

  • Line 81 – should say <50% predicted (or of predicted value, something like that)

We provided an updated description with reference to the predicted values

  • Nice thorough report of key historical factors, including NSAID use
  • Line 89 – may be more appropriate to say “and” rather than “associated with” for the troponin and eos

We apologize for the incorrect sentence. We corrected the sentence as the Reviewer suggested in

Laboratory tests, performed at Emergency department, showed a mild increase of high sensitivity cardiac troponin (hs-TnI 48 ng/L, 99th percentile upper reference limit 14 ng/L) and increased eosinophil count (2080/mm3, 19.5%, total blood cell count 10.68 x 103/microL).”

  • Line 141 – consider “patient declined” rather than “patients refused”

We thank the reviewer for the suggestion. We corrected the case description as requested.

  • Line 144-148

    • Please state whether the patient achieved initial remission (and how that was defined, ? BVAS 0) while on azathioprine and prednisone (sounds like she remained on 10mg/d prednisone).

As the reviewer suggests, the patient achieved remission while on low dose prednisone and azathioprine. We updated the manuscript as requested.

    • If going to report BVAS score, need to state which version you are using.
    •  

We thank the reviewer for the question. We used BVAS  score, version 3, available at:

Mukhtyar C, Lee R, Brown D, et al. Modification and validation of the Birmingham Vasculitis Activity Score (version 3). Ann Rheum Dis. 2009;68(12):1827-1832. doi:10.1136/ard.2008.101279

    • Great that she achieved remission with introduction of mepolizumab. Please comment on duration of follow-up at the time of manuscript submission.

At the time of submission, the patient did not report any relapse while on therapy, after 4 months follow-up

    • Are there any follow-up imaging studies available to determine whether the MRI or PET findings improved after initiation of therapy? This would certainly be interesting to the audience.

We thank the reviewer for the interesting comment. Unfortunately, the patient was managed by a Multidisciplinary team and, according to patient’s preference, she has been referred to a local Rheumatology clinic. Therefore, follow-up cardiovascular imaging studies are not available at the time of submission

DISCUSSION

  • Line 218 – there is an incomplete sentence starting with “Recently, benralizumab”

We apologize for the incomplete sentence. We added a paragraph about the role of benralizumab in the treatment of EGPA-associated myocarditis in the discussion section.

AUTHOR CONTRIBUTIONS, IRB STATEMENT, INFORMED CONSENT STATEMENT – these sections have not been completed by the authors

We apologize with the Reviewer for the inconvenience. We modified the Authors’ contribution section and updated a new version of the manuscript.

FIGURE 1 – phrasing/sequence of the caption could be clearer. May be clearer to state which image is being described prior to the description.

We thank the reviewer for the suggestion. We provided an updated figure description.

FIGURE 2 – Consider saying “small” instead of “mild”

We thank the reviewer for the suggestion. We modified the figure’s description as requested.

Round 2

Reviewer 1 Report

The paper has been improved after the last suggested revisions. In my opinion, it is suitable for publication in the present form.

This manuscript is a resubmission of an earlier submission. The following is a list of the peer review reports and author responses from that submission.